# Magnitude of Hepatitis B virus and associated factors among chronic liver disease patients admitted to the medical ward at Sheikh Hassen Yabare Comprehensive Specialized Hospital, Jigjiga, Ethiopia: a retrospective study

Ramadan Budul Yusuf[1]*, Getnet Addisu[2], Ahmed Mohammed Ibrahim[1,3,4], Terefe Gebre[2], Seid Muhumed Abdilaahi[5], Bilan Ali Ahmed[6], Abdulahi Hussen Abdinur[2], Girma Tadesse Wedajo[1], Mohamed Omar Osman[1]

1 Department of Public Health, Institute of Health Science, Jigjiga University, Jigjiga, Ethiopia, 2 School of Medicine, Institute of Health Science, Jigjiga University, Jigjiga, Ethiopia, 3 Swiss Tropical and Public Health Institute, Kreuzstrasse 2, Allschwil, Switzerland, 4 University of Basel, Petersplatz 1, Basel, Switzerland, 5 Department of Pediatrics and child health Nursing, Institute of Health Science, Jigjiga University, Jigjiga, Ethiopia, 6 Department of Midwifery, Institute of Health Science, Jigjiga University, Jigjiga, Ethiopia

* ramadanbudul83@gmail.com, ramadan.budul@jju.edu.et

## Abstract

### Background

Viral hepatitis is still a major public health problem around the globe, and acquiring adequate and recent epidemiological data on Hepatitis B is important in the prevention and control of the disease. Data on Hepatitis B virus (HBV) among patients with chronic liver disease are scarce in the study area, as well as in Ethiopia at large.

### Objective

To assess the magnitude of Hepatitis B virus (HBV) and its associated factors among chronic liver disease patients admitted to the medical ward at Sheikh Hassen Yabare Comprehensive Specialized Hospital, Jigjiga, Ethiopia.

### Method

An institution-based retrospective cross-sectional study was conducted in the Sheikh Hassen Yabare Comprehensive Specialized Hospital. A total of 249 study participants were selected by simple random sampling using a computer-generated method. A rapid test of HbsAg using a kit from Henso Medical (HANGZHOU) Co. Ltd. was used. The data were collected using Kobo toolbox and analyzed using SPSS version 29.0. Binary logistic regression was employed to identify statistically significant factors for HBV. Model fitness was checked by the Hosmer-Lemeshow goodness-of-fit test, and

**Data availability statement:** All relevant data are within the paper and its Supporting Information files.

**Funding:** The author(s) received no specific funding for this work.

**Competing interests:** The authors have declared that no competing interests exist.

multicollinearity across variables was checked using the variance inflation factor. Statistical significance was considered at P < 0.05, and an adjusted odds ratio with a 95% confidence interval was used to measure the strength of the association.

## Result

The magnitude of HBV among clinically diagnosed chronic liver disease patients was 81/249 (32.53%) (95% CI = 26.9–39.0). Having a history of hospital admission (AOR = 3.60, 95% CI = 1.84–7.04), blood transfusion history (AOR = 2.22, 95% CI = 1.13–4.37), being male (AOR = 2.65, 95% CI = 1.18–5.96) and being exposed to a jaundiced individuals (AOR = 4.52, 95% CI = 2.34–8.72) were also significantly associated with HBV.

## Conclusion

The magnitude of HBV among clinically diagnosed chronic liver disease patients was high. Being male, having a history of hospital admission, blood transfusion, and contact history with jaundice patients were significantly associated with HBV. The hospital should strictly follow blood transfusion safety protocols and implement proper infection control measures to minimize HBV transmission. Additionally, healthcare providers should screen patients with a history of hospital admission, contact with jaundice patients to facilitate early diagnosis and treatment.

## Introduction

A liver inflammation is generally referred to as hepatitis. It happens as a result of autoimmune illnesses, alcoholism, drug exposure, chemical exposure, and toxin exposure, among other pathogen infections [1]. Through immunological anergy, the hepatotropic virus known as the hepatitis B virus (HBV) can cause a chronic and persistent infection in humans [2].

Covalently closed circular DNA (cccDNA), which is arranged into a minichromosome and serves as a template for the production of viral mRNA, is formed during the viral life cycle [3]. Chronic infection and a low rate of recovery are brought on by the extremely stable properties of cccDNA [4]. Depending on the buildup of viral products, an infection might be directly cytopathic, non-cytopathic, or a combination of both in a particular patient [5].

HBV infections can range from asymptomatic infections or moderate illnesses to severe or infrequently fatal fulminant hepatitis. They can also be acute or chronic [6]. Persistent HBV infection, which is defined as the presence of detectable hepatitis B surface antigen (HBsAg) in the blood or serum for more than six months, with or without concurrent active viral replication and signs of hepatocellular injury and inflammation, is a spectrum of diseases. One important determinant of the risk of persistent infection is age. Following an acute infection, chronicity is widespread (90%) in neonates and young children under five years old (20–60%), while it is rare (<5%) in adults [7].

One of the main causes of liver illness is still the hepatitis B virus (HBV), which also raises the risk of late-stage complications such as cirrhosis, hepatocellular carcinoma, and chronic hepatitis [8]. Recently, significant progress has been achieved in the development of a limited HBV treatment that results in the permanent elimination of HBV DNA and HBsAg (a cure), which may offer the means of eradicating HBV as a threat to public health [9]. Significant advancements in the efficacy of therapy have been made in the last few years due to a major improvement in our understanding of the epidemiology and natural history of HBV infection [10]. In terms of reducing HCC incidence, decompensation, and mortality, as well as enhancing histology and inhibiting viral replication, the NA has demonstrated a strong safety profile and excellent efficacy [11].

The Global Health Sector Strategy on Viral Hepatitis was established by the World Health Assembly in 2016 to eradicate the disease by 2030 [12]. A growing body of molecular, clinical, and financial evidence indicates that early antiviral treatment initiation could prevent HCC and save many lives at a very low cost [13]. cccDNA is seldom eliminated by current antiviral treatments [4].

Establishing a functional cure, which is characterized by either the disappearance of hepatitis B surface antigen, undetectable serum HBV DNA levels, or seroconversion to hepatitis B surface antibody, has been the focus of recent antiviral treatment trials for chronic hepatitis B [14]. By ensuring that everyone has equitable access to safe, cost-effective, and preventive diagnosis, treatment, and care of the highest caliber, FMOH hopes to eradicate viral hepatitis as a public health hazard in Ethiopia [15].

In close coordination with the Federal Ministry of Health (FMoH), the EthNoHep group initiated an HBV scale-up treatment program in Ethiopia. To begin treatment, it is based on four factors. Clinically confirmed cirrhosis, APRI >0·7, ALT > 40 U/L, and an HBV viral load >2000 IU/mL are among these requirements. HCC in first-degree relatives is also required [16].

Globally, infections with the hepatitis B virus are a leading cause of infectious disease mortality and have become a significant public health concern [17]. The burden of Hepatitis B virus (HBV) among chronic liver disease patients can vary significantly based on geographical location, prevalence of HBV in the general population, vaccination rates, and access to healthcare [1].

Globally, more than 2 billion individuals are currently infected with HBV, and 350 million of them are chronic carriers of the virus who have a high risk of dying from primary hepatocellular carcinoma, cirrhosis, and active hepatitis [18]. The sub-Saharan region has one of the highest rates of hepatitis B surface antigen (HBsAg) carriers in the world, with a general population prevalence of 5–20% [19]. In this region, the burden of HBV among chronic liver disease patients is likely to be substantial [20].

The prevalence of HBV among CLD patients significantly varies across regions in sub-Saharan Africa [21]. The magnitude was 10.6% in Mbale Regional Referral Hospital, Uganda [22], 42.9% in Ghana [23], and 71% in Nigeria [24]. Ethiopia falls among the epidemiological category of regions where the prevalence of hepatitis B infection is classified as hyper endemic, with a prevalence ranging from 8 to 12% [25]. Evidence in Ethiopia revealed that the magnitude of HBV among CLD patients ranges from 34.8 to 79.5% [26,27].

Various sociodemographic, medical, surgical, obstetric, cultural, and behavioral factors were associated with an increased risk of HBV infection among chronic liver death patients. Some of the factors were middle age, male sex, marriage, living in a rural area, having fewer educational qualifications, smoking, having HBsAg-positive household contacts, having a family history of HBV, having undergone surgery, having received blood transfusions, dental operations in the past and traditional medical procedures [26–29].

The majority (95%) of HBV infections in immunocompetent adults are self-limited and resolve without treatment [6]. However, the virus may cause cirrhosis, and liver cancer, which are the major causes of death, morbidity, and financial hardship [30]. Evidence revealed that the pooled prevalence of cirrhosis was 4.1% in the primary care setting or general population and 12.7% in referral or tertiary care facilities in sub-Saharan Africa regions [31].

Worldwide, the most common cause of cirrhosis and liver cancer is chronic hepatitis B virus infection [32]. In 2019, an estimated 331,000 people died from HBV-related cirrhosis, and around 192,000 people died from HBV-related liver cancer [32]. In 2019, an estimated 331,000 people died from HBV-related cirrhosis, with another 192,000 dying from HBV-related liver cancer 32. Between 1990 and 2019, HBV-related mortality rose by 5.9%, and 2.9% between 2015 and 2019 [33]. According to a Global Burden of Disease study in 2010, cirrhosis mortality increased in sub-Saharan Africa between 1980 and 2010 and it was primarily due to hepatitis B (34%) [34].

The progression of patients with chronic hepatitis B to cirrhosis and HCC is influenced by various factors such as age over 40, male gender, presence of cirrhosis, family history of HCC, Asian and African racial backgrounds, genetic diversity, high levels of HBV replication during follow-up, HBV genotype, HDV, HCV, HIV concurrent infections, heavy alcohol consumption, aflatoxin, smoking, diabetes, and obesity [35].

Evidence in Ethiopia revealed that around 36.7% of chronic liver diseases were attributed to chronic hepatitis B infections [36]. Other evidence showed that HBV is the most common cause of chronic liver disease, accounting for 40% of cases [37] Chronic liver disease is responsible for 12% of hospital admissions and 31% of fatalities in medical wards in Ethiopian hospitals [37].

With prioritized targets, interventions, and innovations, the World Health Organization intends to eradicate chronic viral hepatitis B as a hazard to public health by 2030 [38]. Global plans to eliminate HBV infection include expanding population-level testing and treatment, stepping up HBV prevention interventions, and implementing universal HBV vaccination [39]. The most effective method of eradicating illnesses linked to HBV is vaccination-based HBV infection prevention. A combination of passive immunization with hepatitis B immunoglobulin (HBIG) and active immunization with the HBV vaccine provides the best efficacy in preventing HBV infection [40].

Understanding the magnitude of the Hepatitis B virus and its associated factors among chronic liver disease patients is very important in assessing the burden of the disease and developing effective prevention and control strategies in Ethiopia. However, data on the magnitude of HBV and its associated factors among chronic liver disease patients is scarce in the study area as well as in Ethiopia at large. Therefore, this study was intended to assess the magnitude of HBV and its associated factors among chronic liver disease patients admitted to the medical ward at Sheikh Hassen Yabare Comprehensive Specialized Hospital, Jigjiga, Ethiopia.

## Materials and methods

### Study Area

The study was conducted at the Sheikh Hassen Yabare Comprehensive Specialized Hospital, which is the first and largest specialized teaching hospital in the Somali Region State of Ethiopia It is at the center of Jigjiga city, which is located 610 km away from Addis Ababa in the eastern part of Ethiopia, and it lies at an altitude of 5361 feet (1634 meters) above sea level. It is expected to cover an estimated population of more than seven million people living in the region and neighboring districts of the Oromia region. The hospital also receives referrals from large parts of neighboring countries, like Somaliland and Puntland. The hospital provides health services at the inpatient and outpatient levels as a referral hospital for more than 7 million populations in the eastern part of the country. Under the Department of Internal Medicine, it has general medical wards with a total of 81 beds and chronic follow-up [41].

### Study design and period

Hospital-based cross-sectional study design was conducted using a retrospective medical chart review. The study was conducted from May 01, 2024, to May 30, 2024.

### Source population

All chronic liver disease patients who had received medical care at Sheikh Hassen Yabare Comprehensive Specialized Hospital were the source population.

## Study population

All selected clinically diagnosed chronic liver disease patients who had received medical care at Sheikh Hassen Yabare Comprehensive Specialized Hospital from January 01, 2020, to December 01, 2024 were the study population.

## Inclusion criteria

All chronic liver disease patients aged ≥ 18 years at the time of diagnosis or treatment and who had received medical care at Sheikh Hassen Yabare Comprehensive Specialized Hospital from January 01, 2020, to December 01, 2024.

## Exclusion criteria

All patient charts with incomplete documentation of HBV status were excluded. In addition, medical charts having more than 20% of incomplete variables were excluded.

## Sample size Determination

The sample size was calculated using a single population proportion (p) among chronic liver disease patients.

$$n = \frac{\left(Z_{\frac{\alpha}{2}}\right)2 \times P\,(1-P)}{d^2}$$

Where n = sample population for n>10,000
P = prevalence of HBV surface antigen on CLD(20.4%) [42]
d = marginal error (0.05)
Z (α/2) = the reliability coefficient of 95% i.e. 1.96
By using this formula $n = ((1.96)^2 \times (0.204)(1 - 0.204))/(0.05)^2$, the calculated sample size was 249

## Sampling technique and procedure

Medical record numbers (MRNs) of CLD patients were collected from January 01, 2020, to December 01, 2024, from the Health Management Information System (HMIS) logbook of Sheikh Hassen Yabare Comprehensive Specialized Hospital. A total of 386 Medical record numbers (MRNs) were collected from January 01, 2020, to December 01, 2024. A representative sample was selected by a simple random sampling method using computer generated methods.

## Data collection method

The data collection tool was adapted from previous studies done in Ethiopia [26,27]. The tool contains sociodemographic characteristics, medical-related variables, surgical variables, obstetric and reproductive-related variables, and cultural and behavioral characteristics. First, the tool was entered into Kobo Toolbox software for data collection by the principal investigator.

Then, one-day training was provided for four medical interns (data collectors) by the principal investigator. Medical record numbers (MRNs) of all chronic liver disease patients who had received medical care at Sheikh Hassen Yabare Comprehensive Specialized Hospital from January 01, 2020, to December 01, 2024 were retrieved from the Health Management Information System (HMIS) log Book of Sheikh Hassen Yabare Comprehensive Specialized Hospital. Then, medical charts were selected by the computer-generated random method and drawn from the medical chart office. The Sheik Hassen Yabare Comprehensive Specialized Hospital's main laboratory tested the HBsAg using a rapid test kit Henso Medical (HANGZHOU) Co. Ltd; in accordance with manufacturer protocol and standards, a venous blood sample was collected, and serum was separated for testing, a few drops of serum were applied to the sample well of the test

device, followed by the addition of the assay buffer, the test was then allowed to develop at room temperature for the recommended time (typically 15–20 minutes), the appearance of both a control line and a test line indicated a positive result for HBsAg, while the appearance of only control line indicated a negative result. Tests without a visible control line were considered invalid and repeated. The safety and specimen handling precautions were properly followed during the tests.

Finally, the data collectors collected the data using the Kobo Toolbox software. The investigator managed the data collection procedure and evaluated the tool's completeness using the Kobo Toolbox software.

## Study variables

### Dependent variables

- Hepatitis B virus sero-status (positive/negative)

### Independent variables

- Sociodemographic variable: Age, sex, marital status, occupational status, residence

- Microsurgical variable: history of hospital admission, blood transfusion history, history of surgical procedure, dental extraction

- Cultural and behavioral variables; ear piercing, nose piercing, contact with jaundiced patients,

- sexual and obstetric variables: multiple sexual partners, abortion, sexually transmitted disease

### Operational definition

Hepatitis B virus serostatus can be seropositive or negative. Sero-positive HBV is defined as a positive HBsAg test, whereas Sero-negative HBV is defined as a negative HBsAg test [7].

**Multiple sexual partners**: individuals who have more than one sex partner [43]

Contact history for a jaundiced person: a person who has a history of contact with a known HBsg-positive individual [44].

### Data Quality Control

To ensure the quality of data, one day of training was provided for four medical interns (data collectors) by the principal investigator on the objective of the study, the content of the questionnaire, and techniques of data collection. Before the actual data collection, a pre-test was conducted on 5% of the sample (12 cards) at Karamara Hospital. The collected data were checked continuously for completeness and consistency by the principal investigator daily.

### Data analysis and processing

The data were collected using the Kobo toolbox, for validation and cleaning, and exported into SPSS version 29.0 for analysis, using custom syntax for data re-coding and diagnostics. The study participants' sociodemographic and clinical variables were compiled using descriptive analysis. While continuous variables were summarized using measures of central tendency and dispersion, depending on the data distribution, categorical variables were presented using frequencies and percentages. For clarity, tables and figures were used to present the results. Binary logistic regression was employed to identify statistically significant factors for HBV. In binary logistic regression, both bi-variable and multivariable analyses were done. Variables with a p-value of <0.25 in bi-variable logistic regression were selected as candidate variables for the final model (multivariable logistic regression model). Model fitness was checked by the Hosmer-Lemeshow goodness of test (p-value = 0.224), and multi-collinearity between the explanatory variables was checked using the variance inflation factor (VIF > 10). In multivariable logistic regression, the backward stepwise method was used to select significant variables, and statistical significance was considered at p < 0.05. An adjusted odd ratio (AOR) with a 95% confidence interval was used to measure the strength of the association.

### Ethical considerations

Ethical approval for the research was obtained from the Jigjiga University Research and Ethics Review Board (JJURERB0056/2024) who waived the informed consent due to the retrospective nature of the study. Permission was acquired from the hospitals to extract data from patient medical records; all the patients' data were anonymized and the data from the medical records is held with strong Confidentiality, the Helsinki declaration and the national health ethics guidelines were adhered.

## Results

### Sociodemographic characteristics of study participants

This study included a total of 249 study participants. The age of study participants ranges from 18 to 80 years, with a median age and standard deviation (SD) of 38 ± 15.85 years. Nearly one-third of the study participants (31.7%) were in the age group of 28–37 years. Around two-thirds, 176 (70.7%) of study participants were male. Nearly half (49.8%) of study participants were from urban areas (Table 1).

### Medical, surgical, and culturally related characteristics of study participants

Regarding medical-related characteristics, nearly one-third (32.9%) of study participants had a history of hospital admission. About one-fifth of the study participants (52, 20.9%) had a history of dental extraction. Around one-third (32.5%) of the study participants had a history of blood transfusion. Concerning culturally related characteristics, about 36 (14.5%) of study participants had a history of ear-piercing practices, while around 5 (2%) of study participants had a history of nose-piercing practices. Approximately 105 (42.2%) of study participants had contact history with jaundice patients (Table 2).

### Sexual and obstetric-related characteristics of study participants

Regarding obstetric characteristics, approximately one-fourth (19, or 25.67%) of women had a history of abortion. The majority (92%) of study participants had no history of sexually transmitted infections (STIs), and about 36 (14.5%) of study participants had a history of multiple sexual partners (Table 3).

### Magnitude of HBV among clinically diagnosed chronic liver disease patients

The magnitude of HBV among clinically diagnosed chronic liver disease patients was 81/249 (32.53%) (95%CI = 26.9–38.6%) (Fig 1).

**Table 1. Sociodemographic characteristics of chronic liver disease patients who had received medical care at Sheik Hassen Yabare Comprehensive Specialized Hospital from January 01, 2020, to December 01, 2024.**

| Characteristics | Category | Frequency (n) | Percent (%) |
|---|---|---|---|
| Age (in years) | 18-27 | 45 | 18.1 |
| | 28-37 | 79 | 31.7 |
| | 38-47 | 48 | 19.3 |
| | 48-57 | 31 | 12.4 |
| | ≥58 | 46 | 18.5 |
| Sex | Male | 176 | 70.7 |
| | Female | 73 | 29.3 |
| Residency | Urban | 124 | 49.8 |
| | Rural | 125 | 50.2 |

**Table 2. Medical, surgical, and cultural-related characteristics of chronic liver disease patients who had received medical care at Sheik Hassen Yabare Comprehensive Specialized Hospital from January 01, 2020, to December 01, 2024.**

| Variables | Category | Frequency (n) | Percent (%) |
|---|---|---|---|
| Hospital admission history | Yes | 82 | 32.9 |
| | No | 167 | 67.1 |
| History of blood transfusion | Yes | 81 | 32.5 |
| | No | 168 | 67.5 |
| History of dental extraction | Yes | 52 | 20.9 |
| | No | 197 | 79.1 |
| History of surgical procedure | Yes | 45 | 18.1 |
| | No | 204 | 81.9 |
| History of ear piercing practice | Yes | 36 | 14.5 |
| | No | 213 | 85.5 |
| History of nose piercing practice | Yes | 5 | 2 |
| | No | 244 | 98 |
| Contact history of jaundiced person | Yes | 105 | 42.2 |
| | No | 144 | 57.8 |

**Table 3. Sexual and obstetric-related characteristics of chronic liver disease patients who had received medical care at Sheik Hassen Yabare Comprehensive Specialized Hospital from January 01, 2020, to December 01, 2024.**

| Characteristics | Category | Frequency(n) | Percent (%) |
|---|---|---|---|
| History of abortion (n = 73) | Yes | 19 | 25.67 |
| | No | 54 | 73.97 |
| History of STI | Yes | 20 | 8 |
| | No | 229 | 92 |
| Having multiple sexual partners | Yes | 36 | 14.5 |
| | No | 213 | 85.5 |

## Factors associated with HBV among clinically diagnosed chronic liver disease patients

Binary logistic regression was used to identify statistically significant factors associated with HBV. Bivariable logistic regression was done to identify candidate variables for multiple logistic regressions. Variables such as having a history of hospital admission, dental extraction, surgery, blood transfusion, contact history with jaundice patients, sex, and residency were significant at a p-value of <0.25 in bivariable logistic regression. In multivariable logistic regression, the backward stepwise method was used to select significant variables. Finally, four variables, namely male sex, having a history of hospital admission, blood transfusion, and contact history with jaundice patients, were significantly associated with HBV in clinically diagnosed chronic liver disease patients.

Chronic liver disease patients who had a history of hospital admission were 3.60 times more likely to have HBV infection as compared to their counterparts (AOR = 3.60, 95% CI = 1.84–7.04). In addition, chronic liver disease patients who had a history of blood transfusion were 2.22 times more likely to develop HBV as compared to their counterparts (AOR = 2.22, 95% CI = 1.13–4.37). Furthermore, the likelihood of developing HBV was 2.65 times higher in male chronic liver disease patients as compared to female patients (AOR = 2.65, 95% CI = 1.18–5.96). Moreover, chronic liver disease patients who had a history of contact with patients with jaundice signs and symptoms were 4.52 times more likely to develop HBV as compared to their counterparts (AOR = 4.52, 95% CI = 2.34–8.72) (Table 4).

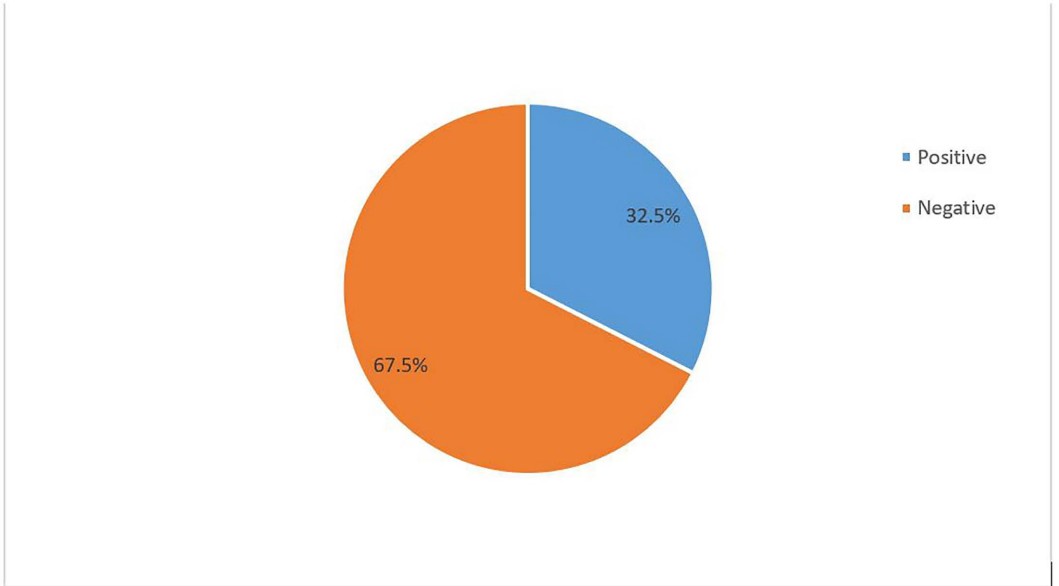

**Fig 1. Magnitude of HBV among clinically diagnosed chronic liver disease patients who had received medical care at Sheik Hassen Yabare Comprehensive Specialized Hospital in 2024.**

**Table 4. Shows the results of multivariable logistic regression for the association between HBV infection and independent variable among chronic liver disease patients who had received medical care at Sheik Hassen Yabare Comprehensive Specialized Hospital from January 01, 2020, to December 01, 2024.**

| Variable | Category | HBV | | COR (95%CI) | AOR(95%CI) |
|---|---|---|---|---|---|
| | | **Positive** | **Negative** | | |
| Sex | Male | 70(86.4) | 106(63.1%) | 3.72(1.83,7.56) | 2.65(1.18, 5.96)* |
| | Female | 11(13.6%) | 62(36.9%) | 1 | 1 |
| History of dental extraction | Yes | 27(33.3%) | 25(14.9%) | 2.86(1.52,5.35) | 1.89(0.87, 4.10) |
| | No | 54(66.7%) | 143(85.1%) | 1 | 1 |
| History of hospital admission | Yes | 49(60.5%) | 33(19.6%) | 6.26(3.48,11.25) | 3.60(1.84, 7.04)* |
| | No | 32(39.5%) | 135(80.4%) | 1 | 1 |
| History of blood transfusion | Yes | 43(53.1%) | 38(22.6%) | 3.87(2.19,6.82) | 2.22(1.13,4.37)* |
| | No | 38(46.9%) | 130(77.4%) | 1 | 1 |
| Contact history of Jaundice | Yes | 59 (72.8%) | 46(27.4%) | 7.11(3.92,12.90) | 4.52(2.34, 8.72)* |
| | No | 22 (27.2%) | 122(72.6%) | 1 | 1 |
| Residency | Urban | 36(44.4%) | 88(52.4%) | 1 | 1 |
| | Rural | 45(55.6%) | 80(47.6%) | 1.37(0.80,2.34) | 1.91(0.99,3.70) |
| History of surgery | Yes | 19(23.5%) | 26(15.5%) | 1.67(0.86,3.24) | 1.62(0.70,3.72) |
| | No | 62(76.5%) | 142(84.5%) | 1 | 1 |

* Indicates statistically significant in multivariable logistic regression with p-value <0.05

## Discussion

The magnitude of HBV among clinically diagnosed chronic liver disease patients was 81/249 (32.53%) (95%CI = 26.9–38.6%), who have distinct socio-cultural practices, healthcare access challenges, and epidemiological patterns. This

finding was in line with a study finding in Iran (38%) [45], the University of Gondar Comprehensive Specialized Hospital (34.8%) [26], and Public Hospitals in Addis Ababa (35.8%) [46].

The findings of this study were lower than those of the study findings in South India (57%) [47], Albania (70%) [48], Ghana (42.9%) [23], and Nigeria (50% and 71% of HBV in liver cirrhosis and hepatocellular carcinoma patients, respectively) [24]. The variation might be because of differences in healthcare access and diagnostic capabilities between the two regions, which may lead to variations in the identification and reporting of HBV cases among chronic liver disease patients. In addition, there may be genetic variations in susceptibility to HBV infection or the progression of HBV-related liver disease between populations. Furthermore, the difference might be due to variations in study periods. Currently, more attention is given to HBV in Ethiopia, and there is an increase in HBV vaccination coverage. This condition may decrease the magnitude of HBV among CLD patients as compared to the previous studies in Albania, Nigeria, and India [49–51].

The finding of this study was also lower than a study finding in Hawassa University Comprehensive Specialized Hospital (79.5%) [27]. The disparity across studies could be explained by differences in regional awareness, availability of preventative services, and health-seeking behaviors between Jigjiga and Hawassa, which can influence who is tested and when. Together, these characteristics indicate that the higher prevalence in Hawassa may be related to more risk-driven screening, whereas the lower rate in Jigjiga indicates a broader but less risk-concentrated sample.

On the other hand, this finding was higher than study findings in the Government Medical College & Rajindra Hospital, India (26%) [52], Yemen (24.1%) [53], and in Mbale Regional Referral Hospital, Uganda (10.6%) [22]. This may be due to the smaller sample sizes in the previous studies, which were 100, 108, and 104 in India, Yemen, and Uganda, respectively, as compared to this study (n = 249). In addition, disparities in healthcare infrastructure, including access to screening, diagnosis, and treatment for HBV, could contribute to variations in prevalence rates. Ethiopia may have limited resources and healthcare facilities compared to Uganda, leading to a higher burden of undiagnosed and untreated HBV cases. Furthermore, variations in cultural practices, beliefs, awareness of the routes of HBV transmission, efforts made by health professionals to implement universal precautions, and preliminary benefits due to the initiation of national immunization programs in other countries could contribute to differences in prevalence rates between Ethiopia and previous studies in Yemen and Uganda.

In this study, chronic liver disease patients with a history of blood transfusion were more likely to develop HBV infection as compared to their counterparts (AOR = 2.22, 95%CI = 1.13–4.37). This finding was supported by a study in Accra, Ghana [52] and Hawassa University Comprehensive Specialized Hospital [27]. The association could be explained by the fact that patients who received blood transfusions many years ago may have been exposed to HBV at a time when screening and testing were not as rigorous as they are today. As a result, they may have developed chronic liver disease over time due to the long-term effects of HBV infection acquired through blood transfusion.

This study revealed that having a history of hospital admission was significantly associated with HBV infection among chronic liver disease patients (AOR = 3.60, 95% CI = 1.84–7.04). This might be due to nosocomial transmission. Hospital settings can be a source of HBV transmission due to the potential for exposure to contaminated blood or bodily fluids. Patients with chronic liver disease who have been hospitalized multiple times may have had increased opportunities for exposure to HBV within healthcare facilities. In addition, patients with chronic liver disease often require frequent medical interventions, such as surgeries, endoscopies, or other invasive procedures. These procedures can pose a risk of HBV transmission if proper infection control measures are not followed, leading to an increased likelihood of acquiring the virus during hospital admissions. Furthermore, in some cases, medical equipment may not be adequately sterilized between uses, increasing the risk of cross-contamination and HBV transmission among patients.

This study also found that having a contact history with jaundiced patients was significantly associated with HBV infection (AOR = 4.52, 95%CI = 2.34–8.72). This finding was in agreement with a study finding in the University of Gondar Comprehensive Specialized Hospital [26]. This could be justified by the direct transmission of the virus from patients with

jaundice signs and symptoms. Jaundice is a common symptom of acute HBV infection, and individuals with jaundice are likely to have high levels of the virus in their blood, making them more infectious.

Finally, this study demonstrated that male sex was significantly associated with HBV infection (AOR = 2.65, 95% CI = 1.18–5.96). This finding was supported by studies conducted in the Taiwanese population [54], Turkey [55], and the University of Gondar Comprehensive Specialized Hospital [26]. This could be explained by high-risk behaviors in males as compared to females. Men are more likely to engage in high-risk behaviors that increase their risk of HBV transmission, such as unprotected sex and intravenous drug use. These behaviors can lead to a higher rate of HBV infection among male individuals. In addition, men may be more likely to work in occupations with a higher risk of exposure to HBV, which increases their risk of infection.

## Limitations of the study

The nature of the retrospective data is prone to missing data, and this could be the limitation of the study. In addition to that, this study did not assess some important behavioral and cultural factors such as the history of uvuloctomy, parenteral drug abuse, history of tattooing, delivery history by traditional birth attendants, traditional body piercing, and sharing sharp instruments. In addition, this study was conducted in a single center and included a small number of study subjects, which may affect its generalizability. The sensitivity and specificity of a rapid test for HbsAg may be lower than those of the laboratory gold standard. Furthermore, chronic liver disease patients with occult HBV infection were not identified due to the retrospective design; in addition to that, patients were not routinely screened at admission, and HBV status at admission was not known, and reverse causality could not be ruled out.

## Conclusion

The magnitude of HBV among chronic liver disease patients was high in Sheik Hassen Yabare Comprehensive Specialized Hospital. Being male, having a history of hospital admission, blood transfusion, and having a contact history with jaundice patients were significantly associated with HBV in patients with chronic liver disease. Hospitals should follow strict blood transfusion safety protocols and implement infection control measures to reduce HBV transmission. Screening patients with a history of hospital admission, blood transfusion, or jaundice contact can facilitate early diagnosis and treatment. Increase public awareness about HBV transmission modes through mass media and campaigns. Future studies should evaluate unaddressed behavioral and cultural factors using a multi-center site, a large number of participants, and a prospective design.

## Supporting information

**S1 File.**
(XLSX)

**S2 File. Annex.**
(DOCX)

## Acknowledgments

The authors express their gratitude to everybody who contributed to this original article at any step.

## Author contributions

**Conceptualization:** Ramadan Budul Yusuf, Getnet Addisu, Ahmed Mohammed Ibrahim, Bilan Ali Ahmed.

**Data curation:** Ramadan Budul Yusuf, Getnet Addisu, Ahmed Mohammed Ibrahim, Terefe Gebre, Seid Muhumed Abdilaahi, Bilan Ali Ahmed, Abdulahi Hussen Abdinur, Girma Tadesse Wedajo, Mohamed Omar Osman.

**Formal analysis:** Ramadan Budul Yusuf, Getnet Addisu, Ahmed Mohammed Ibrahim, Seid Muhumed Abdilaahi, Bilan Ali Ahmed, Abdulahi Hussen Abdinur, Girma Tadesse Wedajo, Mohamed Omar Osman.

**Investigation:** Ramadan Budul Yusuf, Getnet Addisu, Ahmed Mohammed Ibrahim, Terefe Gebre, Seid Muhumed Abdilaahi, Bilan Ali Ahmed, Abdulahi Hussen Abdinur, Girma Tadesse Wedajo, Mohamed Omar Osman.

**Methodology:** Ramadan Budul Yusuf, Getnet Addisu, Ahmed Mohammed Ibrahim, Seid Muhumed Abdilaahi, Bilan Ali Ahmed, Abdulahi Hussen Abdinur, Girma Tadesse Wedajo, Mohamed Omar Osman.

**Project administration:** Ramadan Budul Yusuf, Getnet Addisu, Ahmed Mohammed Ibrahim, Terefe Gebre, Abdulahi Hussen Abdinur, Girma Tadesse Wedajo.

**Resources:** Ramadan Budul Yusuf, Getnet Addisu, Ahmed Mohammed Ibrahim, Bilan Ali Ahmed, Abdulahi Hussen Abdinur, Girma Tadesse Wedajo.

**Software:** Ramadan Budul Yusuf, Getnet Addisu, Ahmed Mohammed Ibrahim, Abdulahi Hussen Abdinur, Girma Tadesse Wedajo, Mohamed Omar Osman.

**Supervision:** Ramadan Budul Yusuf, Getnet Addisu, Ahmed Mohammed Ibrahim, Terefe Gebre, Bilan Ali Ahmed, Abdulahi Hussen Abdinur.

**Validation:** Ramadan Budul Yusuf, Getnet Addisu, Ahmed Mohammed Ibrahim, Terefe Gebre, Abdulahi Hussen Abdinur.

**Visualization:** Ramadan Budul Yusuf, Getnet Addisu, Ahmed Mohammed Ibrahim, Terefe Gebre, Bilan Ali Ahmed, Abdulahi Hussen Abdinur, Girma Tadesse Wedajo.

**Writing – original draft:** Ramadan Budul Yusuf, Getnet Addisu, Ahmed Mohammed Ibrahim, Terefe Gebre, Seid Muhumed Abdilaahi, Bilan Ali Ahmed.

**Writing – review & editing:** Ramadan Budul Yusuf, Getnet Addisu, Ahmed Mohammed Ibrahim, Terefe Gebre, Seid Muhumed Abdilaahi, Abdulahi Hussen Abdinur.

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
