## [Decision Letter · Decision Letter 0]

4 Aug 2025

Dear Dr. Yusuf,

Thank you for submitting your manuscript to PLOS ONE. After careful consideration, we feel that it has merit but does not fully meet PLOS ONE’s publication criteria as it currently stands. Therefore, we invite you to submit a revised version of the manuscript that addresses the points raised during the review process.

We look forward to receiving your revised manuscript.

Kind regards,

Tebelay Dilnessa, MSc

Academic Editor

PLOS ONE

Journal Requirements:

Additional Editor Comments:

Title should be revised as: Magnitude of Hepatitis B virus and associated factors among chronic liver disease patients admitted to the medical ward at Sheikh Hassen Yabare Comprehensive Specialized Hospital, Jigjiga, Ethiopia: a retrospective studyA thorough revision of the English language is needed. The writing lacks fluency and has serious issues with grammar that need to be addressed. The document suffers from major coherence problems. It requires an intensive revision in all aspects.HBV: It should be written in both forms at least once as ‘Hepatitis B virus (HBV)’Lines 33 &34: A total of 249 study subjects were selected by…… It should be written as; A total of 249 study participants was included……………….Which laboratory technique/method was employed to diagnose HBV? It should be mentioned both in the abstract and methods part.In the abstract and result, the absolute number (numerator and denominator) is needed together with the percentage. For example, A/B (C%).Your conclusion in the abstract part should be supported by evidence. It was beyond your scope and requires revision.Line 153: Materials and methodsThe laboratory method should be clearly stated in the methods part even though retrospective.Line 166: Study design and periodThe inclusion and exclusion criteria should be placed separately and clearly.Line 182: The sample size for the first specific objective was calculated using a single population…Line 184: The formula should be prepared by word text, not figure.Discussion not merely comparison, but it includes justifications of your resultsYour conclusion and recommendation were too ambitious. It should be specific and based on your results.Figures and tables should be prepared to the standard.The tables can be inserted where you cited within the main manuscript/text.Unnecessary bolds should be avoided from the ‘conclusion’ part. The recommendation seems specific to concerned body. Please write in a paragraph form.The references were not written properly. All references should be written correctly according to Vancouver style.Follow the standard binomial nomenclature, italize journal name and the word ‘et al’NB: follow the guideline for manuscript writing protocol for PLoS One.

Reviewers' comments:

Reviewer's Responses to Questions

**Comments to the Author**

1. Is the manuscript technically sound, and do the data support the conclusions?

Reviewer #1: Yes

Reviewer #2: Yes

2. Has the statistical analysis been performed appropriately and rigorously?

Reviewer #1: Yes

Reviewer #2: I Don't Know

3. Have the authors made all data underlying the findings in their manuscript fully available?

Reviewer #1: No

Reviewer #2: No

4. Is the manuscript presented in an intelligible fashion and written in standard English?

Reviewer #1: Yes

Reviewer #2: Yes

Reviewer #1: The methods section, while comprehensively detailing the statistical analyses and ethical considerations, could benefit from a deeper elaboration on certain aspects to enhance transparency and reproducibility. Specifically, while a "lottery method" for simple random sampling is mentioned, more granular detail on its practical execution could be provided to ensure clarity for replication. Additionally, given the retrospective nature of the study, a brief acknowledgment within the methods about the inherent limitations of using pre-existing medical records, such as potential data inconsistencies or missing information, could add a layer of robustness. Finally, while software versions are cited, further specifics on the implementation within tools like Kobo Toolbox (e.g., form validation, data cleaning protocols) or custom scripts used in SPSS, beyond standard regression techniques, would enrich the methodological description.

Reviewer #2: Line 186: The formula uses a prevalence rate (P) of 20.4%, but no source is cited. Please add a reference.Lines 304-306: The claim that the current result (32.53%) is lower than the Hawassa study (79.5%) is attributed to "increased vaccination coverage." However, no local vaccination data is provided to support this inference. It is recommended to add a citation referencing literature on Ethiopia's hepatitis B vaccination rate.

**Do you want your identity to be public for this peer review?** For information about this choice, including consent withdrawal, please see our Privacy Policy

Reviewer #1: **Yes: ** Abdukadir Nigus Mohammed

Reviewer #2: No

---

## [Author Response · Author response to Decision Letter 1]

21 Aug 2025

Reviewers' comments:

Comments to the Author

Reviewer #1:

The methods section, while comprehensively detailing the statistical analyses and ethical considerations, could benefit from a deeper elaboration on certain aspects to enhance transparency and reproducibility. Specifically, while a "lottery method" for simple random sampling is mentioned, more granular detail on its practical execution could be provided to ensure clarity for replication. Additionally, given the retrospective nature of the study, a brief acknowledgment within the methods about the inherent limitations of using pre-existing medical records, such as potential data inconsistencies or missing information, could add a layer of robustness. Finally, while software versions are cited, further specifics on the implementation within tools like Kobo Toolbox (e.g., form validation, data cleaning protocols) or custom scripts used in SPSS, beyond standard regression techniques, would enrich the methodological description.

Reviewer's Responses:

Dear reviewer, thank you for your comments and suggestions:

• We revised the ‘lottery method’ into computer computer-generated method to select the study participants in both the abstract and method sections.

• We added that the retrospective nature of the records can lead to missing data, and we incorporated it in the limitation section of the manuscript.

• We used Kobo Toolbox for the validation and cleaning of the data

• We used custom syntax for data re-coding and diagnostics

Reviewer #2: Line 186: The formula uses a prevalence rate (P) of 20.4%, but no source is cited. Please add a reference. Lines 304-306: The claim that the current result (32.53%) is lower than the Hawassa study (79.5%) is attributed to "increased vaccination coverage." However, no local vaccination data is provided to support this inference. It is recommended to add a citation referencing literature on Ethiopia's hepatitis B vaccination rate.

Reviewer's Responses:

Dear reviewer, we are grateful to get these insightful comments and suggestions:

• We have cited the study from which we took the proportion to calculate the sample size.

• We have cited the paragraphs on the lines you quoted

• We have revised the justification for the discussion section based on your question and suggestion.

Editor’s comments

Title should be revised as: Magnitude of Hepatitis B virus and associated factors among chronic liver disease patients admitted to the medical ward at Sheikh Hassen Yabare Comprehensive Specialized Hospital, Jigjiga, Ethiopia: a retrospective study

• Thanks, dear editor, we have revised per your suggestion

A thorough revision of the English language is needed. The writing lacks fluency and has serious issues with grammar that need to be addressed. The document suffers from major coherence problems. It requires an intensive revision in all aspects.

• We revised as per your recommendation

HBV: It should be written in both forms at least once as ‘Hepatitis B virus (HBV)’

• We revised based on your recommendation

Lines 33 &34: A total of 249 study subjects were selected by…… It should be written as; A total of 249 study participants was included……………….

• Dear editor, we have revised based the suggestion

Which laboratory technique/method was employed to diagnose HBV? It should be mentioned both in the abstract and the methods part.

• We have incorporated the method of laboratory investigation of the participants.

In the abstract and result, the absolute number (numerator and denominator) is needed together with the percentage. For example, A/B (C %).

• Thanks for the input, we have included the numbers as well as the percentage in the abstract and the results

Your conclusion in the abstract part should be supported by evidence. It was beyond your scope and requires revision.

• We have revised based on the main findings

The laboratory method should be clearly stated in the methods part, even though retrospective

• We have mentioned the laboratory methods both in the abstract and the method section

Line 166: Study design and period

• Dear editor, we have reshuffled based on the suggestion

The inclusion and exclusion criteria should be placed separately and clearly.

• We have separated the inclusion and the exclusion criteria based on the given suggestion

Line 182: The sample size for the first specific objective was calculated using a single population…

• We have revised based on the suggestion

Line 184: The formula should be prepared by word text, not figure.

• We prepared it in text form and revised accordingly

Discussion not merely a comparison, but it includes justifications of your results

• We have revised based on your comments and suggestions

Your conclusion and recommendation were too ambitious. It should be specific and based on your results.

• We revised it based on the main findings

Figures and tables should be prepared to the standard.

• We have revised based on the given suggestion

The tables can be inserted where you cited within the main manuscript/text.

• We acknowledge your suggestions and inserted the tables into the main manuscript/text

Unnecessary bolds should be avoided from the ‘conclusion’ part. The recommendation seems specific to concerned body. Please write in a paragraph form.

• We appreciate your concerns, and we have amended based on the suggestions

The references were not written properly. All references should be written correctly according to Vancouver style

• We made all the references Vancouver style

Follow the standard binomial nomenclature, italize journal name and the word ‘et al’

• Dear editor, we have revised based on the suggestion.

NB: follow the guidelines for manuscript writing protocol for PLoS One.

• Well noted

---

## [Decision Letter · Decision Letter 1]

8 Sep 2025

Dear Dr. Yusuf,

Thank you for submitting your manuscript to PLOS ONE. After careful consideration, we feel that it has merit but does not fully meet PLOS ONE’s publication criteria as it currently stands. Therefore, we invite you to submit a revised version of the manuscript that addresses the points raised during the review process.

We look forward to receiving your revised manuscript.

Kind regards,

Tebelay Dilnessa, MSc

Academic Editor

PLOS ONE

Journal Requirements:

**Additional Editor Comments:**

- Please stick to the PLOS ONE manuscript writing protocol and write using standard language, as well proofread.

- The P-value should be included in the table than indicating as legend.

Reviewers' comments:

Reviewer's Responses to Questions

**Comments to the Author**

Reviewer #1: All comments have been addressed

Reviewer #2: All comments have been addressed

2. Is the manuscript technically sound, and do the data support the conclusions?

Reviewer #1: Partly

Reviewer #2: Yes

3. Has the statistical analysis been performed appropriately and rigorously?

Reviewer #1: Yes

Reviewer #2: I Don't Know

4. Have the authors made all data underlying the findings in their manuscript fully available?

Reviewer #1: Yes

Reviewer #2: No

5. Is the manuscript presented in an intelligible fashion and written in standard English?

Reviewer #1: Yes

Reviewer #2: Yes

Reviewer #1: This manuscript reports original research on the prevalence and determinants of hepatitis B virus infection among chronic liver disease patients in Ethiopia. The study addresses an important public health issue in a region where HBV is endemic, and the findings are clearly presented with descriptive statistics and logistic regression analysis. The identification of significant risk factors, including hospital admission, blood transfusion, male sex, and contact with jaundiced individuals, provides valuable insights that could inform local prevention and screening strategies. The work is relevant and falls within the scope of PLOS ONE.

However, several areas need improvement before the manuscript can be considered for publication. The methods require greater detail on sample selection, diagnostic procedures, and handling of missing data to ensure reproducibility. The manuscript also lacks a complete ethics approval statement and does not comply with PLOS ONE’s strict data availability policy; the dataset must be deposited in a public repository. In addition, the contribution of this study compared to previous Ethiopian work should be more clearly articulated, tables and references should be reformatted to meet journal style, and the language edited for clarity. Overall, the article has merit but requires minor revision to meet the journal’s standards.

Reviewer #2: The authors are commended for addressing the previous comments with additional citations and explanations. However, upon careful review of the revised manuscript, I find that several points still require modification and clarification.

1.The study relies on a rapid HBsAg test kit for HBV diagnosis. The potential for lower sensitivity and specificity of such kits compared to laboratory gold standards could lead to an underestimation of the true disease prevalence. This question should be acknowledged and discussed in the 'Limitations' section.

2.The manuscript concludes that hospitalization is a risk factor for HBV infection. However, this interpretation overlooks the potential for reverse causality. It is plausible that severe HBV-related liver disease itself necessitates frequent hospital admissions. This explanation should be included in the Discussion.

3.The manuscript lacks consistency in the presentation of statistical notations. For instance, confidence intervals are variously formatted as "95%CI" and "95CI" across the text. Similarly, the notation for p-values is not uniform. These terms must be standardized throughout the manuscript for clarity and professionalism.

**Do you want your identity to be public for this peer review?** For information about this choice, including consent withdrawal, please see our Privacy Policy

Reviewer #1: **Yes: ** Abdukadir Nigus Mohammed

Reviewer #2: No

---

## [Author Response · Author response to Decision Letter 2]

24 Oct 2025

Reviewers' comments:

Comments to the Author

Reviewer #1: The methods require greater detail on sample selection, diagnostic procedures, and handling of missing data to ensure reproducibility. The manuscript also lacks a complete ethics approval statement and does not comply with PLOS ONE’s strict data availability policy; the dataset must be deposited in a public repository. In addition, the contribution of this study compared to previous Ethiopian work should be more clearly articulated, tables and references should be reformatted to meet journal style, and the language should be edited for clarity. Overall, the article has merit but requires minor revision to meet the journal’s standards.

Response to Reviewer

Dear reviewer, we acknowledge your insight.

• We have put the details of the diagnostic procedure

• We have mentioned in the exclusion criteria that incomplete records were excluded, and the issue of missing data was handled in that manner.

• We have also deposited the data into a public repository (Zenodo), and it can be accessed at (https://doi.org/10.5281/zenodo.17264843).

• The ethical approval statement was provided, and the detail was mentioned under the ethical consideration section.

• The tables and the figures were formatted to meet the journal’s style

• All the formatting and grammatical issues were addressed.

Reviewer #2:

1. The study relies on a rapid HBsAg test kit for HBV diagnosis. The potential for lower sensitivity and specificity of such kits compared to laboratory gold standards could lead to an underestimation of the true disease prevalence. This question should be acknowledged and discussed in the 'Limitations' section.

Response to Reviewer

Dear reviewer, we highly appreciate your insight.

We have addressed and included the potential for lower sensitivity and specificity compared to the gold standard in the limitations section.

2. The manuscript concludes that hospitalization is a risk factor for HBV infection. However, this interpretation overlooks the potential for reverse causality. It is plausible that severe HBV-related liver disease itself necessitates frequent hospital admissions. This explanation should be included in the discussion.

Response to Reviewer

Dear reviewer, We highly appreciate your constructive insight. Since there were no HBV routine screenings on patients at admission, the issue of reverse causality could not be ruled out, and we have addressed it under the limitation section, with a recommendation of a prospective study in the future.

3. The manuscript lacks consistency in the presentation of statistical notations. For instance, confidence intervals are variously formatted as "95%CI" and "95CI" across the text. Similarly, the notation for p-values is not uniform. These terms must be standardized throughout the manuscript for clarity and professionalism.

Response to Reviewer

Dear reviewer, we highly appreciate your constructive insight. We have adjusted all the statistical notations per the comment, and all the terms were consistently written with clarity.

Editor’s comments

1. Expand the Methods section and include Ethics approval details.

• Thanks, Dear Editor, we have expanded the method section, including the diagnostic procedure and ethical approval details

2. Deposit the dataset in a public repository with a DOI.

• We have also deposited the data into a public repository (Zenodo), and it can be accessed at (https://doi.org/10.5281/zenodo.17264843).

3. Strengthen the justification of novelty.

• Dear editor, we appreciate your concern; this study was conducted in a population with distinct socio-cultural practices of pastoralism, healthcare access challenges, and epidemiological patterns.

4. Revise references, tables, and figures to match journal style.

• All are revised to match the journal style

5. Proofread the text for clarity and grammar.

• Thanks, we addressed all the formatting and grammatical issues

---

## [Decision Letter · Decision Letter 2]

2 Nov 2025

Dear Dr. Yusuf,

Thank you for submitting your manuscript to PLOS ONE. After careful consideration, we feel that it has merit but does not fully meet PLOS ONE’s publication criteria as it currently stands. Therefore, we invite you to submit a revised version of the manuscript that addresses the points raised during the review process.

We look forward to receiving your revised manuscript.

Kind regards,

Tebelay Dilnessa, MSc

Academic Editor

PLOS ONE

Journal Requirements:

Additional Editor Comments :

Line 35: (HANGZHOU, Co. Ltd.); similarly do it in the methods partInsert the assumptions and prevalence (P) to the sample size determination formula and show it.Line 170: The study was conducted from May 01, 2024, to May 30, 2024.Write lines 176, 181, 195,197 and 206 similar to line 170 (the date).Line 205: Please write out the full form of MRNs. A sentence cannot start with an abbreviation.Please ensure that all dates in each table are aligned with the above comment.

Reviewers' comments:

Reviewer's Responses to Questions

**Comments to the Author**

Reviewer #1: All comments have been addressed

2. Is the manuscript technically sound, and do the data support the conclusions?

Reviewer #1: Yes

3. Has the statistical analysis been performed appropriately and rigorously?

Reviewer #1: Yes

4. Have the authors made all data underlying the findings in their manuscript fully available?

Reviewer #1: Yes

5. Is the manuscript presented in an intelligible fashion and written in standard English?

Reviewer #1: Yes

Reviewer #1: The revised version of the manuscript shows substantial improvement compared to the previous submission. The authors have carefully and adequately addressed all the comments and suggestions raised during the prior round of review. The clarifications and additional details provided in the methodology, data presentation, and discussion sections have enhanced the scientific quality, clarity, and overall readability of the paper.

The study on the magnitude and associated factors of HBV among clinically diagnosed chronic liver disease patients at Sheik Hassen Yabare Comprehensive Specialized Hospital provides valuable epidemiological insights relevant to both local and regional public health. The findings contribute to understanding the burden of HBV and its relationship with key clinical and behavioral risk factors in an underrepresented population.

The revised manuscript now:

Presents results with improved organization and coherence.

Clearly describes the sampling, data collection, and analytical methods.

Provides well-supported conclusions aligned with the presented data.

Meets the ethical standards required for human subject research and follows proper data confidentiality protocols.

Uses clear, grammatically sound, and intelligible English throughout.

The statistical analyses are appropriate and adequately explained. The interpretation of the adjusted odds ratios is consistent with the data, and the discussion appropriately situates the findings within the context of existing literature. No issues related to dual publication, plagiarism, or research ethics are apparent.

Overall, the manuscript satisfies PLOS ONE’s publication criteria for originality, methodological soundness, ethical integrity, and clarity of reporting. I find the study scientifically valid and relevant to the field of clinical epidemiology and infectious disease research.

**Do you want your identity to be public for this peer review?** For information about this choice, including consent withdrawal, please see our Privacy Policy

Reviewer #1: **Yes: ** Abdukadir Mohammed

---

## [Author Response · Author response to Decision Letter 3]

25 Nov 2025

Editor Comments:

1. Line 35: (HANGZHOU, Co. Ltd.); similarly, do it in the methods part

Author’s response: Thanks, dear editor, we have done so based on your suggestion on line 213.

2. Insert the assumptions and prevalence (P) into the sample size determination formula and show it.

Author’s response: We really appreciate your insightful suggestion, and we inserted the assumptions into the formula, line 193.

3. Line 170: The study was conducted from May 01, 2024, to May 30, 2024. Write lines 176, 181, 195, 197, and 206 similar to line 170 (the date).

Author’s response: Dear editor, we have changed the date to the format you suggested for lines 170, 176, 181, 196, 199, 209, 278, 291, 170, 176, 181, 196, 199, 209, 278, 291, 300, and 328, including the tables.

4. Line 205: Please write out the full form of MRNs. A sentence cannot start with an abbreviation.

Author’s response: Dear editor, we acknowledge your suggestions and have written the full form in all lines starting with MRNs (medical record numbers) for lines 196, 198, and 207.

5. Please ensure that all dates in each table are aligned with the above comment.

Author’s response: We really appreciate your insight and have fully aligned the dates per the comment you have given us.

---

## [Editor Report · Decision Letter 3]

26 Nov 2025

Magnitude of Hepatitis B virus and associated factors among chronic liver disease patients admitted to the medical ward at Sheikh Hassen Yabare Comprehensive Specialized Hospital, Jigjiga, Ethiopia: a retrospective study.

PONE-D-25-31070R3

Dear Dr. Yusuf,

We’re pleased to inform you that your manuscript has been judged scientifically suitable for publication and will be formally accepted for publication once it meets all outstanding technical requirements.

Kind regards,

Tebelay Dilnessa, MSc

Academic Editor

PLOS ONE
---

## [Editor Report · Acceptance letter]

PONE-D-25-31070R3

PLOS One

Dear Dr. Yusuf,

I'm pleased to inform you that your manuscript has been deemed suitable for publication in PLOS One. Congratulations! Your manuscript is now being handed over to our production team.

Kind regards,

on behalf of

Dr. Tebelay Dilnessa

Academic Editor

PLOS One